# Depression Symptom Patterns and Social Correlates among Chinese Americans

**DOI:** 10.3390/brainsci8010016

**Published:** 2018-01-16

**Authors:** Lin Zhu

**Affiliations:** Center for Asian Health, Lewis Katz School of Medicine, Temple University, 3440 N Broad St., Kresge Bldge, Ste. 320, Philadelphia, PA 19140, USA; lin.zhu@temple.edu

**Keywords:** Chinese Americans, depression symptom, social correlates, nativity status, psychiatric symptomatology

## Abstract

The aim of this study is to examine and compare the depression symptoms pattern and social correlates in three groups: foreign-born Chinese Americans, US-born Chinese Americans, and non-Hispanic whites. This study used data from the Collaborative Psychiatric Epidemiology Surveys (CPES). The study sample consists of 599 Chinese Americans (468 for the foreign-born and 121 for the US-born) and 4032 non-Hispanic whites. Factor analysis was used to examine the depression symptom patterns by each subgroup. Four depression symptoms dimensions were examined: negative affect, somatic symptoms, cognitive symptoms, and suicidality. Logistic regression was used to investigate the effects of sociodemographic (age, gender, marital status, and education), physical health condition, and social relational factors (supports from and conflict with family and friends) on specific types of depression symptoms separately for the three subgroups. The findings showed little differences in depression symptom patterns but clear variation in the social correlates to the four depression dimensions across the three ethnocultural groups, foreign-born Chinese Americans, US-born Chinese Americans, and non-Hispanic whites. Clinicians should take into account the sociocultural factors of patients when making diagnosis and suggesting treatments. In addition, psychiatrists, psychologists, or other mental health service providers should offer treatment and coping suggestions based on the specific symptom dimensions of patients, and patients’ ethnocultural backgrounds.

## 1. Introduction

To date, most studies of depression among Chinese Americans have focused on comparing the prevalence of depression to other racial/ethnic groups [1,2], or identifying risk factors among Chinese Americans [3,4,5]. In comparison, there are fewer academic studies examining the depression symptom patterns of Chinese Americans. Cultural psychiatrists such as Arthur Kleinman [6,7] and Anthony Marsella [8] have asserted that Chinese in China or Taiwan, as well as Chinese Americans, express their depression or impaired mood in a distinct way, and they attribute such cultural differences in depression symptomatology to the collectivism and the philosophical orientation in traditional Chinese culture. Specifically, some scholars pointed out that people with Chinese cultural backgrounds tend to express more somatic symptoms and less affective or emotional symptoms [7,9,10,11].

Several elements in Chinese culture, including the orientations towards collectivism, and the philosophical orientation towards holism between body and mind, man and nature, are distinct from the mainstream American culture, which is primarily Western. Additionally, these distinct cultural elements are associated with the construction, reflection, and expression of self, which are key factors in the cause and expression of impaired mood [12]. Is the pattern of depressive symptoms among Chinese Americans different from that among non-Hispanic whites? The answer to this question is the first step towards more thorough understanding of the etiology and pathophysiology of depression. 

Depression is a multidimensional condition, with symptoms manifested in different aspects [13,14]. Specifically, measurements of depression include screening tests like the Center for Epidemiologic Studies Depression Scale-Revised (CESD-R) [13] or clinical assessment criteria like the Diagnostic and Statistical Manual of Mental Disorders (DSM). Most measures contain symptoms of various dimensions: emotional/affective symptoms (e.g., feeling blue, discouraged, happy, felt like achieved a lot), somatic symptoms (e.g., fatigue, insomnia), interpersonal relation function (e.g., feeling others are unfriendly, feeling critical of others), self-perception or existential symptoms (feeling hollow, does not respect self), and suicidality (e.g., thought about death, feeling it would be better if dead). However, most studies of the depression did not consider the multidimensionality of depression [15]; they examined either a binary outcome (depressed or not) [16] or a unidimensional outcome of the total number of symptoms [17]. Therefore, their findings might obscure any differences in depressive symptomatology between groups.

This study seeks to explore the patterns of social correlates foreign-born Chinese Americans, US-born Chinese Americans, and non-Hispanic whites, and to examine the social correlates of depressive symptoms separately for each of the three groups. Through cross-group comparisons, this study hopes to advance our understanding of the sociocultural aspects expression depression symptoms.

### 1.1. Depression Symptomatology among Chinese Americans

Epidemiological literature has inconsistent findings on whether Chinese Americans are less prone to depression than non-Hispanic whites. Some studies found a higher risk of depression among Chinese Americans than among non-Hispanic whites, after covariates are controlled for, and researchers argued that such disparities were due to racial discrimination, acculturative stress, or lower use of mental health services [2,18]. In contrast, studies have reported a lower level of depression among Chinese Americans than non-Hispanic whites or predominantly white samples [1,4], and scholars attributed the lower depression risk among Chinese Americans to strong social support, family cohesion, and other sociocultural factors [19]. Meanwhile, other scholars hypothesized that the difference in depression prevalence between Chinese or Chinese Americans and non-Hispanic whites is partially due to differences in depressive symptom patterns. Cultural factors have a significant influence on the way people experience and express impaired mood [6,8]. 

#### 1.1.1. Somatic Symptoms

Scholars have noted a tendency among the Chinese populations to somatize their psychological distress [9,10]. The term *somatization* is used to refer to the reporting of (usually) medically unexplained symptoms by patients with psychiatric disorders [20]. Some scholars argued that the reporting of somatic symptoms is actually the denial of psychological distress, i.e., that somatic symptoms serve as “a psychological defense against the awareness or expression of psychological distress” [20]. Other researchers pointed out that the higher level of somatic symptoms reported by Chinese people, compared to those by Euro-Canadians, were likely due to a tendency toward externally oriented thinking, rather than identifying emotions or describing them [21]. Their analysis suggested that somatic symptoms are noticed and reported more by individuals who do not focus much on their internal feelings and emotional states. Additionally, since externally oriented thinking, an aspect of alexithymia, was a preferred thinking style in Chinese culture, it could potentially explain the cultural differences in the prevalence somatic symptoms.

Although epidemiological studies have found somatization in various cultural groups, relevant theoretical work indicates that the somatization tendency is particularly strong among Chinese individuals. The health beliefs in ancient Chinese medicine are based on the *yin*/*yang* cosmology. The human body is perceived as a mixture of *yin* and *yang*, and the balances of the two forces lead to health. An imbalance of *yin* and *yang*, i.e., a dissonance of man and nature, would result in illness, which is rarely localized, but affects the entire human being [22]. These philosophical understandings lead to the conception of a very close connection between the mind and the body, which is different from the conceptions of the human body based on “absolute dichotomies and unresolvable differences” [23]. 

#### 1.1.2. Affective Symptoms

Affective symptoms of depression often include pessimism, dissatisfaction, crying, and irritability [24,25]. Research also found that Chinese or Chinese Americans suppress the expression of emotional/affective symptoms [11], or that Chinese Americans tend to express their emotional disturbances in somatic terms [26]. Chinese culture values self-control and endurance, and discourages emotional expression, especially verbal expression [27]. Depression as an illness carries strong social stigma; it is still poorly understood in Chinese society, and is sometimes equated to lunacy and schizophrenia [28]. Therefore, to “save face” for the family and the individuals, individuals may tend to hide their emotional/affective disturbance. One study [29] found that depressed European Americans showed deceased positive affect or emotional reactions (i.e., smiles, happiness) compared to those not depressed; in contrast, depressed Asian Americans showed similar or even greater affect or emotional reactions than those not depressed. Their findings supported the argument that cultural differences in emotional expression norms affect the emotional/affective responses in depressed individuals between cultural groups.

Previous findings on the depressive symptom patterns of Chinese or Chinese Americans are inconsistent. Some studies found no difference in the prevalence of affective symptoms between Chinese and Western populations [20,30]. In contrast, Okazaki [31] found a higher level affective distress among Asian American adults than that among white Americans. Additionally, one study found that Hong Kong Chinese reported more somatic symptoms, and less affective symptoms than did Americans [32]. 

#### 1.1.3. Other Depression Dimensions

Cognitive patterns and interpersonal relationships are also influenced by cultural forms and psychosocial orientations [33,34,35], and depression may affect the expression of these symptoms differently for individuals with different cultural backgrounds. Only a few studies have reported any difference of other dimensions of depression between Chinese or Chinese Americans and other groups, with limited and inconsistent findings. One Australian study found that the Malaysian Chinese sample significantly less likely than the Caucasian Australian sample to report cognitive symptoms [36]. In contrast, other researchers found that adolescents in Hong Kong with subthreshold major depressive disorder (MDD) showed similar social function and cognitive symptoms with US adolescents with MDD [37]. With regards to interpersonal symptoms, Chentsova-Dutton and Tsai [38] found that interpersonal complaint plays an important role in the conceptions of depression; Asian individuals, including Chinese, emphasize more heavily interpersonal stressors than do Western cultural groups. In addition, a US study found that Asian Americans reported higher levels of social anxiety than did white Americans [31]. These limited evidences suggested potentially distinctive patterns of cognitive and interpersonal symptoms among Chinese Americans.

### 1.2. Social Correlates of Depression Dimensions among Chinese Americans

Various demographic and social factors influence a multiplicity of events and experiences, which then affect the expression of depression symptoms [39]. However, our knowledge is rather limited on the effects of the sociodemographic factors on specific dimensions of depression in the general US population or Asian Americans. Furthermore, previous studies suggested notable heterogeneity of depression prevalence and predictors by ethnicity among Asian Americans [40]. Literature on Chinese Americans is scarce. The theories of development of psychiatric symptomatology [41,42] have suggested that four types of factors serve as social precursors of depressive symptoms, including (1) demographic and socioeconomic (e.g., age, gender, race, educational attainment), (2) vulnerability factors (e.g., chronic physical condition), (3) protective factors (e.g., social support), and (4) provoking factors (e.g., social conflict). Empirical studies have also examined the effects of these factors on various dimensions of depression symptoms, but mostly among exclusively or predominately non-Hispanic white populations. Below I summarize previous findings by each type of social precursors.

#### 1.2.1. Demographic and Socioeconomic Factors

Previous studies have found some difference in the prevalence of certain dimensions of depression symptoms by gender, age, educational attainment, employment status, and nativity status. First, regarding the gender difference in depression symptomatology, two studies have found that the female preponderance in depression rates is more evident (and consistent in the literature) in negative or positive affect and weight gain than it is other dimensions of symptoms [15,43]. Kessler and colleagues [44] also found that an older age was associated with more somatic symptoms for both men and women; however, an older age was associated with more affective symptoms for only women and not men. Their findings suggest that the age and gender impact different dimensions of depression symptoms through nuanced mechanisms. Educational attainment and employment are life-achievement variables that previous research suggests has effects on depression [45,46,47]. However, these studies only examined depression as one whole disease category, and failed to explore any differences in the socioeconomic effects on different dimensions of depression symptoms. 

Previous literature also suggests that nativity status is an important factor for the depression symptomatology of immigrants. For example, one study found that suicidal ideation was more prevalent in US-born Asian Americans than that among the foreign-born Asian immigrants. Additionally, suicidal ideation varied between men and women, but only among the US-born, and not among the foreign-born [46]. Furthermore, there are clear variations in the level of acculturation, subscription into the host culture, and retention of the heritage culture, between foreign-born and native-born immigrants [48], and the process of negotiating between the host and the heritage culture has a complex impact on psychological outcomes [49]. Therefore, it would be easier to dissect the complex social effects on depression symptomatology if we were to examine the foreign-born separately from the US-born. More specifically, this study explores how several sociodemographic factors, including age, gender, marital status, educational attainment, and employment status, are associated with each depressive symptom dimension, separately for foreign-born Chinese Americans and US-born Chinese Americans, as well as for non-Hispanic whites.

#### 1.2.2. Vulnerability Factors

The presence of any chronic physical conditions may cause not only functional disability, but also tremendous psychological distress [50], evidenced in the high co-morbidity rates of depression and numerous chronic illness or conditions such as chronic pain, diabetes, or cancer [51,52,53]. In addition, previous research has found the effects of chronic physical conditions across all dimensions of depression [54,55]. However, given the cultural influences on the expression of depression symptoms, it is necessary to examine if chronic physical conditions exert effects equally on each dimensions of depressive symptoms. In the present study, I explore the effects of having any chronic physical conditions on each depressive symptom dimension, separately for foreign-born Chinese Americans and US-born Chinese Americans, as well as for non-Hispanic whites.

#### 1.2.3. Protective Factors

Although social support is generally perceived to have positive effects on the psychological well-being of individuals, the effects are nuanced. Not only do the sources (family, friends, work relationships, etc.) and forms (emotional, instrumental, etc.) of social support matter, it has also been suggested that the protective effects of social support may vary across different dimensions of depressive symptoms. For example, Blazer and colleagues [56] found that although a higher level of social support was associated with more somatic symptoms and more negative affect, the effect was almost twice for somatic symptoms as it was for negative affect symptoms. More specifically, regarding Chinese Americans, no study has specifically examined the effects of social support across different depression dimensions. Studies on suicidal ideation consistently found significant social support effect among Chinese or Chinese Americans [57,58]. These findings suggest differential effects of social support on various depression symptom dimensions. 

There are several important cultural values and beliefs in Chinese culture that link to social support and the expression of depression. For example, Chinese culture values social harmony over individual’s well-being, but at the same time, it emphasizes group cohesion and the interdependency among individuals. How the complex cultural matrix may affect the specific forms of depression expression is not yet clear. In the present study, I explore how perceived supports from friends and from relatives are associated with each depressive symptom dimension, separately for foreign-born Chinese Americans and US-born Chinese Americans, as well as for non-Hispanic whites.

#### 1.2.4. Provoking Factors

The distress caused by interpersonal relationships among family and peers plays a major role in affecting an individual’s psychosocial vulnerability [3,59]. Most studies found that a higher level of social conflict was associated with a higher depression risk among Chinese Americans [5,60]. For example, Lepore [61] found that a lower level of conflicts with friends was associated with better emotional functioning, and the association was as strong as the effects of social support among predominantly white college students in the US. There is a richer literature on the association between greater social conflict and a higher level of suicidal ideation among Chinese or Chinese Americans [62]. 

To summarize, the existing body of literature shed some light on the social correlates of depression dimensions among Chinese Americans, but two major issues are yet to be addressed. First, it is unclear what factors are associated with each specific type of depression dimensions. Second, no study has compared the patterns of social correlates between Chinese Americans and non-Hispanic whites. Therefore, this study hypothesizes, based on previous findings, that foreign-born Chinese Americans and US-born Chinese Americans have depression symptom patterns and social correlates that are distinct from those among non-Hispanic whites. This is a very general hypothesis, which is appropriate for this study, an exploratory effort to better understand how Chinese Americans express depression symptoms.

## 2. Methods

### 2.1. Data Source

This study used publicly available data from the Collaborative Psychiatric Epidemiology Surveys (CPES) funded by the National Institute of Mental Health (NIMH) [63]. This survey joins together three nationally representative surveys: the National Comorbidity Survey Replication (NCS-R), the National Survey of American Life (NSAL), and the National Latino and Asian American Study (NLAAS). The CPES survey population included adults age 18 and older, living in households in the 48 coterminous United States (NCS-R, NSAL), and the population for the Latino and Asian ancestry groups extend to the State of Hawaii (NLAAS). All three surveys were conducted between 2001 and 2003. Respondents in all three surveys were selected from a four-stage clustered area probability sample of households. 

This study focused on Chinese American respondents while using non-Hispanic whites as a comparison group. The CPES data consist of 600 Chinese Americans from NLAAS, and 7587 non-Hispanic whites from NCS-R and NSAL. Specifically, the non-Hispanic white subsample from NSAL (*n* = 891) was not considered optimal for descriptive analysis of the white adult population, or comparative analyses between whites and non-black minority groups [64], and was therefore dropped. Also excluded are 2516 non-Hispanic white cases from NCS-R Part I who did not complete the more in-depth interview. The procedure results in a sample of 4180 non-Hispanic white respondents from NCS-R. The sample used in the factor analysis, after excluding cases with missing values on depression symptoms, consists of 599 Chinese Americans and 4032 non-Hispanic whites. In the analysis for social correlates of depression symptom dimensions, after excluding cases with missing values in all variables entered in the binary logistic regression models, the sample size is 599 for Chinese Americans (468 for the foreign-born and 121 for the US-born) and 3749 for non-Hispanic whites.

### 2.2. Measures

#### 2.2.1. Major Depressive Disorder Symptoms

The diagnostic criteria of MDD in the fourth edition of DSM (DSM-IV) include twenty-four questions that could be combined into indicators of nine specific symptoms: depressed mood, decreased interest or pleasure, change in weight or appetite, change in sleep, change in activity, fatigue, concentration, worthlessness/guilt, and suicidality. Specifically, respondents were asked if they had certain symptoms during a depressive episode, defined as the period of several days or longer during which their sadness or discouragement or lack of interest and other problems were most severe and frequent. The twenty-four questions also could be put into four dimensions, based on existing literature. The four dimensions are: negative affect (seven items of depressed mood, decreased interest or pleasure, and worthlessness/guilt), somatic symptoms (seven items of change in weight or appetite, insomnia or hypersomnia, and fatigue), cognitive symptoms (five items of agitation and retardation, and concentration), and suicidality (five items). 

#### 2.2.2. Race/Ethnicity and Generation Status

Race/ethnicity was defined using the self-identified primary racial/ethnic identity. Three generation groups were identified among Chinese Americans: foreign-born to foreign-born parents (1st generation), US-born to at least one foreign-born parent (2nd generation), and US-born adults to US-born parents (3rd-or-higher generations). 

#### 2.2.3. Chronic Physical Condition

Respondents were asked if they ever had cancer, headaches, heart disease, high blood pressure, asthma, ulcer, or stroke. Respondents who had at least one of the six conditions were coded as “yes”, and those who did not were coded as “no”.

#### 2.2.4. Social Support and Conflicts

Social support from friends and relatives was measured separately, each with three questions. Support from friends was measured on a scale consisting of three parallel items that assess how much respondents can rely on friends for emotional support. Scores were *z*-standardized on the Chinese Americans and non-Hispanic whites combined, with a higher numeric value indicating higher level of friend support. How much respondents can rely on extended family for emotional support was measured on a scale consisting of three items on how often respondents talk on the phone, get together with relatives, and how often they can rely on relatives to discuss worries. Scores were *z*-standardized, with a higher numeric value indicating a higher level of relative support. The answers were “not at all”, “little”, “some”, or “a lot”. In addition, conflict with friends and relatives were assessed separately, each with two questions. Specifically, how often friends or relatives made too many demands of respondents and argue with respondents. The answers were “never”, “rarely”, “some”, or “often”. For friend conflict and relative conflict, the scores were *z*-standardized, and a higher numeric value represented a higher level of conflict. 

#### 2.2.5. Sociodemographic Correlates

In regard to the measurement of race/ethnicity, when a respondent identifies with multiple race/ethnicity categories, he or she was assigned to a single category according to priority order in the NSAL and NLAAS respondent classification rules (e.g., Vietnamese over Chinese). Other sociodemographic correlates include gender, age in years, marital status, having a college degree, work status, and family income level. Specifically, marital status was coded in three categories, never married, married/cohabiting, and divorced/separated/widowed. The family income level was computed as the ratio of the household income to the 2001 US Department of Health and Human Services poverty guideline. 

### 2.3. Statistical Analysis

All statistical analyses were conducted using Stata 14 [65]. Exploratory factor analysis on all twenty-four items of MDD symptoms were used to examine the patterns of depression symptoms for three groups, foreign-born Chinese Americans, US-born Chinese Americans, and non-Hispanic whites. The results of the factor analysis on depressive symptom patterns are presented in this section, because it is essentially a measurement issue. 

#### 2.3.1. Descriptive Statistics of the Twenty-Four Items

Before conducting the factor analysis, I examine the prevalence of all items across the three groups of interest. As Table 1 shows, for all but two items (“weight gain over 10 lbs” and “weight loss over 10 lbs”), the prevalence was lowest among foreign-born Chinese Americans, higher among US-born Chinese Americans, and highest among non-Hispanic whites. For all items but “weight gain over 10 lbs”, the prevalence among foreign-born Chinese Americans is significantly lower than that among non-Hispanic whites. For fourteen of the items, the prevalence among foreign-born Chinese Americans is significantly lower than that among US-born Chinese Americans. Six of the items, “felt depressed”, “discouraged about things in life”, “lost interest in things”, “trouble sleeping”, “low energy and tired without work”, and “more trouble concentrating” are ones with the highest prevalence for all three groups. Five items, “larger appetite than usual”, “slept more than usual”, “other notice restlessness”, “made suicide plan”, and “attempted suicide” are the items with the lowest prevalence for all three groups. 

#### 2.3.2. Factor Analysis

All twenty-four depression symptom items are subject to a common factor analysis separately for each of the three groups. For each group, the eigenvalues of the first ten factors in the factor analysis without rotation are presented in Table 2, along with the proportion of variance explained by each factor. Three factors for foreign-born Chinese Americans, four factors for US-born Chinese Americans, and two factors for non-Hispanic whites have eigenvalues greater than 1.0. However, for all three groups, the eigenvalue drops drastically after the second factor, and the proportions fall below 10%. For that reason, and for clearer comparison among the three groups of interest, I extracted two factors for each group. Oblique oblimin rotation was used, because correlated factors are hypothesized [66]. (The results of factor analysis using orthogonal rotation are available on request). The summaries of factor analysis for all three groups are presented in Table 3. Factor loadings for all items are presented in Table 4, with loadings greater than 0.40 (a general rule of thumb suggests using a cutoff of 0.40 of for factor loadings) are marked in bold. Figure 1 is a visual presentation of the factor loadings from the two factors for the three groups separately.

Among the 468 foreign-born Chinese Americans, Factor A1 accounts for 62 percent of the total variance, with an eigenvalue of 10.79 (Table 2). (The proportions sum to greater than 100 percent, because the factors have overlapping explanatory powers.) Seventeen items, including all seven of the negative affect items, four items of the somatic symptoms (wa1, wa4, ih1, and fa1), four items of the cognitive symptoms (ar1, cc1–3), and two suicidality items (sc1–2), load strongly onto this factor. There is a lack of conceptual unity among these seventeen variables. The second factor is less than half the factor size of Factor A1, explaining only 26% of the total variances, with an eigenvalue of 4.48. Four suicidality items (sc2–4) load strongly onto this factor. Together the two factors explain 88 percent of the total variance among the items. 

Among the 122 US-born Chinese Americans, the two factors explain 80 percent of the total variance. The first factor, Factor B1, has a factor size of 59 percent of the total variance, with an eigenvalue of 11.99. Seventeen items—the same one that loads strongly onto Factor A1—also load strongly onto Factor B1, and one suicidality item sc3. Factor B2 has an eigenvalue of 4.34, and it explains 21 percent of the total variance. Four items (sc2–5) load strongly onto Factor B2. 

Among 4032 non-Hispanic whites, only two factors are extracted. The first one, Factor C1 (78 percent of variation), is strongly correlated with the second one, Factor C2 (36 percent of variation) (*r* = 0.51). Fifteen items, including seven items of negative affect, three somatic symptoms (wa1, ih1, and fa1), four cognitive symptoms (ar1, and cc1–3), and one suicidality item (sc1), load strongly onto Factor C1. Four suicidality items (sc2–5) load strongly onto Factor C2. In summary, Factor C1 accounts for a mixture of negative affect, somatic symptoms, cognitive symptoms, and suicidality, while Factor C2 accounts for suicidality solely. 

By comparing the patterns of the patterns in the three plots of factor loading in Figure 1, the author concludes that there are only minor differences in the factor structures of the three groups. The US-born Chinese Americans seem to present a more clustered factor loading pattern (red diamonds in the middle plot), which indicates a slightly tighter and clearer two-factor structure, especially Factor B1 (the clustered on the right end). One other notable variation is wa2, which loads strongly onto Factor B2 for US-born Chinese Americans, but does not load strongly onto any factor for the other two groups. Overall, the factor analysis shows factor structural similarity across the three groups.

Binary logistic regression was used to examine the social correlates of the four depressive symptom dimensions created from the factor analysis for Chinese Americans. The CPES supplied weighting was used in all analytical procedures to adjust for the study’s complex sampling methods using the “svy” command. Consequently, multivariate significance testing uses Wald $\chi^{2}$ tests based on coefficient variance-covariance matrices that adjusted for design effects using the Taylor’s series linearization method. To examine any differences of sociodemographic characteristics and depression symptom dimensions by the three groups, Rao-Scott chi-square test [67], a design-adjusted version of the Pearson chi-square test was used for categorical factors and weighted linear regression, was used for continuous factors.

## 3. Results

The social correlates of the four dimensions of depression (negative affect, somatic symptoms, cognitive symptoms, and suicidality) were examined. Given the similarity of the factor analyses among the groups and the lack of clear conceptual distinctions among the dimensions, the factor analysis results for the analysis of social correlates were not used; instead, four conceptually-defined dimensions of depression were used for the analysis of social correlates: negative affect, somatic symptoms, cognitive symptoms, and suicidality among Chinese Americans. These are valid and distinctive dimensions, conceptually and practically. They affect quality of life of individuals differently, and call for different treatment and coping mechanisms. Therefore, this study investigated how some of the demographic and social factors are associated with each of the four dimensions among Chinese Americans.

Weighted descriptive statistics of the four depression dimensions and sociodemographic factors for foreign-born Chinese Americans, US-born Chinese Americans, and non-Hispanic whites are presented in Table 5. The presented *p*-values indicate whether the variable differ significantly across the three groups, while the significance of the pairwise comparisons are indicated with superscripts. 

The proportions of people that experience negative affect, somatic symptoms, cognitive symptoms, and suicidality vary significant across three groups. For all four depression dimensions, the proportions among foreign-born Chinese Americans are significantly lower than those among US-born Chinese Americans, and significantly lower than those among non-Hispanic whites. The differences between US-born Chinese Americans and non-Hispanic whites are not statistically significant. 

US-born Chinese Americans have the lowest average age (40.00), which is significantly lower than that of foreign-born Chinese Americans (43.70) and non-Hispanic whites (46.64). Foreign-born Chinese Americans have a much higher proportion (74.68%) of the married or cohabitating, and much a lower proportion of the divorced/separated/widowed, than US-born Chinese Americans (47.20% and 17.20%) and non-Hispanic whites (61.21% and 19.32%). The marital status composition varies significantly across the three groups. Also, foreign-born Chinese Americans (42.72%) and US-born Chinese Americans (50.37%) are significantly more likely to have a college degree than the non-Hispanic whites (26.64%). 

The prevalence of any chronic physical condition is much higher in non-Hispanic whites (49.95%) than it is for foreign-born (26.81%) or US-born (39.03%) Chinese Americans. Foreign-born Chinese Americans have significantly a significantly lower level of friend support (−0.65) and a significant lower level of relative support (–0.60) than do US-born Chinese Americans (0.09 and −0.01, respectively) and non-Hispanic whites (0.09 and 0.10, respectively). US-born Chinese Americans have much higher levels of friend conflict (−0.25) or relative conflict (0.17) than foreign-born Chinese Americans (−0.05 and −0.04, respectively) and non-Hispanic whites (−0.06 and −0.12, respectively). 

The results of the weighted binary logistic regression models among foreign-born Chinese Americans (*n* = 468) are presented in Table 6. In Model 1-1, foreign-born Chinese Americans who have at least one chronic physical condition (β=1.04) are more likely to have any negative affect than those without. A higher level of relative conflict (β=0.47) is also related to a greater chance of having negative affect. Among US-born Chinese Americans who have at least one chronic physical condition, (β=1.13) are more likely to have any somatic symptoms than those without. A higher level of relative conflict (β=0.49) is also associated with a higher probability of having any somatic symptoms. In Model 1-3, having any chronic physical conditions is the only significant predictor, with those that have at least one chronic physical conditions associated with a higher likelihood of having any cognitive symptoms (β=0.99). Model 1-4 does not find any significant predictors for suicidality. 

The results of the weighted binary logistic regression models among US-born Chinese Americans (*n* = 121) are presented in Table 7. Among foreign-born Chinese Americans, women (β=1.83) are more likely than men to have any negative affect. Those with a college degree (β=1.35) are more likely than those without to have any negative affect. In addition, those with any chronic physical conditions (β=1.32) are more likely than those without to have any negative affect. The same three factors are also significant predictors of somatic symptoms among US-born Chinese Americans, in Model 2-2, with identical regression coefficients. Among US-born Chinese Americans, women (β=2.07) are more likely than men to have at least one cognitive symptom. Having a college degree (β=0.98) and having any chronic physical conditions (β=1.27) are associated with more cognitive symptoms. In addition, a lower level of friend support (β=−0.67) is associated with a higher likelihood of having any cognitive symptoms. In Model 2-4, gender is the only significant predictor of suicidality among US-born Chinese Americans, with women (β=1.86) more likely than men to have any suicidality symptoms. 

Table 8 shows the results from the weighted logistic regression of four depressive dimensions on social correlates among non-Hispanic whites. Six factors, age, gender, being married or cohabitating, having chronic physical conditions, relative support, and relative conflict, are significantly associated with all four depression dimensions. In Model 3-1, age (β=0.10) and its quadratic term (β=−0.001) are significantly associated with any negative affect, and the age effect is in a reversed U-shape. Women (β=0.51) are more likely than men to have negative affect. Those who are married or cohabitating (β=−0.42) are less likely than those who are never married to have any negative affect. Those with any chronic physical conditions (β=0.76) are more likely than those without to have any negative affect. A higher level of relative support (β=−0.16) is related to a lower likelihood of having any negative effect, while a higher level of relative conflict (β=0.21) is associated with a higher likelihood of having any negative affect. These six factors are also significantly associated with the other three dimensions, with similar regression coefficients.

## 4. Discussion

In this study, the author explored the patterns of social correlates foreign-born Chinese Americans, US-born Chinese Americans, and non-Hispanic whites, and examined the social correlates of depressive symptoms separately for each of the three groups. Few differences were found in depression symptom patterns, but there were clear variations in the social correlates of the four depression dimensions across the following three ethnocultural groups: foreign-born Chinese Americans, US-born Chinese Americans, and non-Hispanic whites. Factor analysis revealed very little difference in depression symptoms among the three groups. Four of the suicidality items, “thought would be better if dead”, “thought about suicide”, “made suicide plan”, and “attempted suicide” load strongly onto a factor, while the rest load strongly onto another factor. Some researchers distinguish suicidal ideation from depression; instead of treating suicidal ideation as a depression symptom or dimension, they consider the former a result of the latter, and therefore examine depression and other mental disorders as a risk factors [68]. It seems, according to the finding of the present study, that suicidality should be considered as a distinct concept. Given that suicidal ideation and attempts tend to have more serious and immediate life-threatening outcomes, at the very least, sub-analysis should be conducted on this dimension of depression symptoms. 

Although the finding of the present study, namely, very little cultural difference in depression symptom patterns, seems to endorse the reliability of the DSM-IV diagnostic criteria, it is crucial to note the lack of measurement on positive affect and interpersonal relationships. Since these two dimensions of depression have been confirmed to be an expression of impaired mood in different groups [69], not including them in diagnostic criteria or screening tests would be problematic. By not measuring these two dimensions, is DSM-IV underdiagnosing depression among Chinese Americans and other cultural groups? This is a question that future research needs to answer. The diagnostic criteria is slightly different in the fifth version of DSM (DSM-V). One notable change is the bereavement exclusion for a major depressive episode that was applied to depressive symptoms that lasted less than two months following the death of a beloved one [70]. It is worth exploring how the exclusion might affect prevalence of MDD among different ethnocultural groups, since there are culturally diverse bereavement practices [71] and psychological and physical consequences.

Another major finding of this study is that there are differences in the social correlates of the depression dimensions across the following three groups: foreign-born Chinese Americans, US-born Chinese Americans, and non-Hispanic whites. Table 6, Table 7 and Table 8 show clear differences in which factors are significantly associated with one or more depression dimensions. The obvious difference between foreign-born Chinese Americans, US-born Chinese Americans, and non-Hispanic whites in terms of social correlates indicates that clinicians need to take into account the sociocultural factors of patients when making diagnosis and suggesting treatments [72]. Psychiatrists, psychologists, or other mental health service providers should offer treatment and coping suggestions based on the specific symptom dimensions of patients, and patient’s ethnocultural background. For example, if a first-generation Chinese immigrant patient reports a lot of somatic symptoms and at the same time a high level of conflicts with relatives, the psychiatrist should help patients make sense of the conflicts in talk or behavioral therapy, and suggest the patient make it a priority to address the relative conflicts. If a US-born Chinese American patient is showing agitation or other cognitive symptoms, the psychiatrist should encourage greater communication with friends. 

This study is not without limitations. Because of the lack of measurement on positive affect and interpersonal symptoms in DSM-IV diagnostic criteria, I was unable to examine the role of these two dimensions in the entire depressive patterns, or their social correlates. Previous findings suggest significant differences in these two dimensions between cultural groups. In addition, the groups of foreign-born Chinese Americans and US-born Chinese Americans have drastically smaller sample size than that of non-Hispanic whites, which could be the reason that I found much fewer significant predictors for the first two groups than for the non-Hispanic white group. The much larger sample size of non-Hispanic whites could mean much smaller standard error, and hence a coefficient that is not significant for foreign-born or US-born Chinese Americans might be significant for non-Hispanic whites. Furthermore, research is need to examine prevalence, symptom patterns, and social correlates of DSM-V endorsed MDD. As noted before, with the bereavement exclusion no longer applied in DSM-V, it is worthwhile to examine how this change might affect different ethnocultural groups.

Findings of the present study provide suggestions for future research. Specifically, cultural norms regarding depression and social stigma have been theorized and hypothesized to affect the expression of depressive symptoms, yet there have been very few empirical studies that test such effects, largely due to the difficulty in operationalizing stigma and cultural norm. Findings from previous research [73,74] shed light on this topic, which merit future research. 

## 5. Conclusions

In conclusion, the findings of this study found very similar factor structures of DSM-IV depressive symptoms among foreign-born Chinese Americans, US-born Chinese Americans, and non-Hispanic whites. For all three groups, suicidal ideation or attempt is a construct that is distinct from the rest of the symptoms items. The three groups have different social correlates, yet there are only minor differences in the social correlates for each one of the four depression dimensions within each group. Age is significantly associated with all four dimensions for non-Hispanic whites, but not significant for Chinese Americans. The female preponderance is found for all four dimensions among US-born Chinese Americans and non-Hispanic whites. Having college degree is a unique predictor of negative affect, somatic symptoms, and cognitive symptoms, but for US-born Chinese Americans only. Non-Hispanic whites who were married or cohabitating differ significantly from those never married on negative affect, somatic symptoms, and cognitive symptoms.

## Figures and Tables

**Figure 1 brainsci-08-00016-f001:**
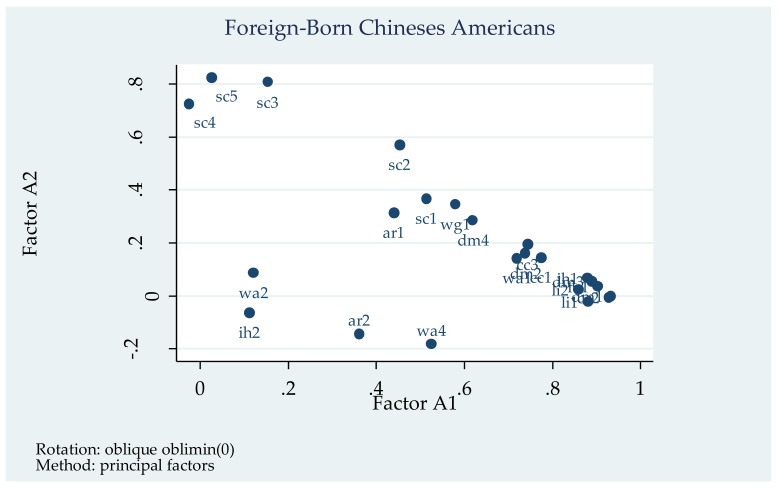
Factor Loading Plot.

**Table 1 brainsci-08-00016-t001:** Nine Depression Symptoms Measured by Twenty-Four Questions in DSM-IV.

	Foreign-Born Chinese Americans	US-Born Chinese Americans	Non-Hispanic Whites
% (*n*)	(468)	(122)	(4032)
Negative Affect			
Felt depressed most days ^α,β,γ^	10.04% (47)	28.86% (34)	35.62% (1436)
Nothing could cheer you most days ^α,β^	5.77% (27)	20.49% (25)	23.29% (939)
Discouraged about things in life most days ^α,β^	9.40% (44)	23.77% (29)	33.28% (1342)
Felt hopeless about future most days ^β^	4.06% (19)	13.93% (17)	22.87% (922)
Lost interest in things you used to think fun ^α,β^	7.26% (34)	22.95% (28)	28.77% (1160)
Nothing fun though good things happening ^α,β^	5.98% (28)	22.13% (27)	26.12% (1053)
Worthlessness or excessive guilt ^α,β^	3.63% (17)	15.57% (19)	15.77% (636)
Somatic Symptoms			
Small appetite most days ^α,β^	6.62% (31)	21.31% (26)	24.55% (990)
Larger appetite than usual most days ^β^	0.43% (2)	2.46% (3)	5.46% (220)
Weight gain ≥ 10 lbs	0	0	0
Weight loss ≥ 10 lbs ^α,β,γ^	1.28% (6)	7.38% (9)	0
Trouble sleeping most nights ^β^	8.97% (42)	19.67% (24)	27.53% (1110)
Slept more than usual most nights ^β^	0.64% (3)	3.28 (4)	6.37% (257)
Low energy and tired without work most days ^α,β^	8.55% (40)	26.23% (32)	31.82% (1283)
Cognitive Symptoms			
Others notice talk/move more slowly ^α,β^	2.90% (14)	13.11% (16)	14.01% (565)
Others notice restlessness ^β^	0.64% (3)	0.82% (1)	3.94% (159)
Slow or mix-up thoughts ^α,β^	6.20% (29)	17.21% (21)	21.25% (857)
More trouble concentrating most days ^α,β^	8.33% (39)	23.77% (29)	29.64% (1195)
Unusual indecisiveness ^β,γ^	5.13% (24)	13.93% (17)	24.11% (972)
Suicidality			
Often thought of death ^α,β^	3.63% (17)	18.03% (22)	21.80% (879)
Would be better if dead ^β^	3.21% (15)	11.48% (14)	16.39% (661)
Thought about suicide ^α,β^	1.71% (8)	9.84% (12)	12.43% (501)
Made suicide plan ^β^	0.43% (2)	2.46% (3)	4.07% (164)
Attempted suicide ^β^	0.85% (4)	1.64% (2)	3.143% (126)

^α^ significant difference between foreign-born and US-born Chinese Americans; ^β^ significant difference between foreign-born Chinese Americans and non-Hispanic white; ^γ^ significant difference between US-born Chinese Americans and non-Hispanic white.

**Table 2 brainsci-08-00016-t002:** Eigenvalues of First Ten Factors from Initial Factor Analysis.

	Foreign-Born Chinese Americans		US-Born Chinese Americans		Non-Hispanic Whites	
	Eigenvalue	Proportion	Eigenvalue	Proportion	Eigenvalue	Proportion
Factor 1	11.37	0.66	12.3	0.60	10.96	0.80
Factor 2	1.99	0.12	2.03	0.10	1.32	0.10
Factor 3	1.25	0.07	1.20	0.06	0.88	0.06
Factor 4	0.96	0.03	1.04	0.05	0.59	0.04
Factor 5	0.50	0.03	0.93	0.05	0.37	0.03
Factor 6	0.43	0.02	0.84	0.04	0.25	0.03
Factor 7	0.34	0.02	0.67	0.03	0.24	0.02
Factor 8	0.29	0.01	0.48	0.02	0.14	0.02
Factor 9	0.26	0.01	0.35	0.02	0.12	0.02
Factor 10	0.22	0.01	0.26	0.01	0.01	−0.02

**Table 3 brainsci-08-00016-t003:** Factor Analysis Summary.

Foreign-Born Chinese Americans			US-Born Chinese Americans			Non-Hispanic Whites		
	Variance	Proportion		Variance	Proportion		Variance	Proportion
Factor 1	10.79	0.62	Factor 1	11.99	0.59	Factor 1	10.80	0.78
Factor 2	4.48	0.26	Factor 2	4.34	0.21	Factor 2	4.98	0.36
Correlation	Factor 1	Factor 2	Correlation	Factor 1	Factor 2	Correlation	Factor 1	
Factor 2	0.29		Factor 2	0.31		Factor 2	0.51	

Principal factor method, oblique oblimin rotation.

**Table 4 brainsci-08-00016-t004:** DSM-IV MDD Symptoms Factor Analysis.

		Foreign-Born Chinese Americans			US-Born Chinese Americans			Non-Hispanic Whites		
Symptoms	#	Factor A1	Factor A2	Unique-ness	Factor B1	Factor B2	Unique-ness	Factor C1	Factor C2	Unique-ness
Negative Affect										
Felt depressed most days	dm1	0.93	−0.01	0.13	0.90	0.04	0.16	0.96	−0.05	0.13
Nothing could cheer you most days	dm2	0.74	0.16	0.36	0.85	0.14	0.18	0.76	0.07	0.36
Discouraged about things in life most days	dm3	0.89	0.06	0.18	0.87	0.08	0.20	0.92	−0.01	0.16
Felt hopeless about future most days	dm4	0.62	0.28	0.44	0.75	0.13	0.37	0.70	0.19	0.35
Lost interest in things you used to think fun	li1	0.88	−0.02	0.24	0.87	0.09	0.18	0.88	−0.01	0.24
Nothing fun though good things happening	li2	0.86	0.02	0.25	0.84	0.09	0.23	0.84	−0.02	0.30
Worthlessness or excessive guilt	wg1	0.58	0.35	0.43	0.73	0.21	0.32	0.53	0.26	0.51
Somatic Symptoms										
Small appetite most days	wa1	0.72	0.14	0.40	0.90	−0.20	0.26	0.77	−0.06	0.46
Larger appetite than usual most days	wa2	0.12	0.09	0.97	−0.003	0.53	0.72	0.28	0.03	0.91
Weight gain >= 10 lbs	wa3	-	-	-	-	-	-	-	-	-
Weight loss >= 10 lbs	wa4	0.52	−0.19	0.75	0.58	−0.09	0.69	-	-	-
Trouble sleeping most nights	ih1	0.88	0.07	0.19	0.81	0.02	0.34	0.84	−0.08	0.36
Slept more than usual most nights	ih2	0.11	−0.06	0.99	0.32	0.09	0.91	0.30	0.01	0.91
Low energy and tired w/out work most days	fa1	0.90	0.03	0.17	0.88	0.02	0.21	0.93	0.004	0.22
Cognitive Symptoms										
Others notice talk/move more slowly	ar1	0.44	0.31	0.63	0.77	0.04	0.38	0.59	0.02	0.65
Others notice restlessness	ar2	0.36	−0.14	0.88	0.22	0.04	0.96	0.23	−0.10	0.94
Slow or mix-up thoughts	cc1	0.77	0.14	0.32	0.86	−0.04	0.27	0.76	−0.01	0.42
More trouble concentrating most days	cc2	0.93	−0.01	0.43	0.88	−0.01	0.24	0.92	−0.09	0.22
Unusual indecisiveness	cc3	0.74	0.19	0.33	0.73	0.10	0.40	0.80	−0.01	0.37
Suicidality										
Often thought of death	sc1	0.51	0.37	0.49	0.71	0.29	0.29	0.55	0.33	
Would be better if dead	sc2	0.45	0.57	0.32	0.41	0.57	0.36	0.36	0.56	0.35
Thought about suicide	sc3	0.15	0.81	0.25	0.42	0.56	0.36	0.19	0.71	0.33
Made suicide plan	sc4	−0.02	0.73	0.48	−0.02	0.83	0.32	0.04	0.71	0.52
Attempted suicide	sc5	0.03	0.82	0.31	−0.09	0.85	0.32	0.07	0.66	0.60

DSM-IV: Diagnostic and Statistical Manual of Mental Disorders, Fifth Edition; MDD: major depressive disorder; Principal factor method, oblique oblimin rotation.

**Table 5 brainsci-08-00016-t005:** Weighted Descriptive Statistics of Sociodemographic Characteristics.

	Foreign-Born Chinese American	US-Born Chinese American	Non-Hispanic White
	*n* = 468	*n* = 121	*n* = 3794
any negative affect ^α,β^	10.06% (0.02)	25.47% (0.04)	25.05% (0.01)
	*p* < 0.001		
any somatic symptoms ^α,β^	9.74% (0.02)	25.47% (0.04)	24.76% (0.01)
	*p* < 0.001		
any cognitive symptoms ^α,β^	8.14% (0.02)	22.21% (0.04)	22.35% (0.01)
	*p* < 0.001		
any suicidality ^α,β^	4.09% (0.01)	14.99% (0.03)	16.64% (0.01)
	*p* < 0.001		
age ^β,γ^	43.70 (1.23)	40.00 (2.09)	46.64 (0.56)
	*p* < 0.05		
gender			
female	54.20% (0.02)	49.83% (0.04)	51.51% (0.01)
male	45.80% (0.02)	50.17% (0.04)	48.49% (0.01)
	not significant		
marital status ^α,β,γ^			
never married	15.53% (0.02)	35.61% (0.07)	19.47% (0.01)
married/cohabitating	74.68% (0.03)	47.20% (0.09)	61.21% (0.01)
divorced/separated/widowed	9.79% (0.02)	17.20% (0.05)	19.32% (0.01)
	*p* < 0.001		
having college degree ^α,β^	42.72% (0.04)	50.37% (0.05)	26.64% (0.01)
	*p* < 0.001		
any chronic conditions ^α,β^	26.81% (0.03)	39.03% (0.05)	49.95% (0.01)
	*p* < 0.001		
friend support (*z*) ^α,β^	−0.65 (0.06)	0.09 (0.08)	0.09 (0.02)
	*p* < 0.001		
relative support (*z*) ^α,β^	−0.60 (0.06)	0.01 (0.12)	0.10 (0.02)
	*p* < 0.001		
friend conflict (*z*) ^β,γ^	−0.05 (0.06)	0.25 (0.08)	−0.06 (0.02)
	*p* < 0.01		
relative conflict (*z*) ^β,γ^	−0.04 (0.05)	0.17 (0.10)	−0.12 (0.02)
	not significant		

^α^ significant difference between foreign-born and US-born Chinese Americans; ^β^ significant difference between foreign-born Chinese Americans and non-Hispanic white; ^γ^ significant difference between US-born Chinese Americans and non-Hispanic white.

**Table 6 brainsci-08-00016-t006:** Weighted Logistic Regression on Four Dimensions of Depression Among Foreign-born Chinese Americans.

	Model 1-1	Model 1-2	Model 1-3	Model 1-4
	Negative Affect	Somatic Symptoms	Cognitive Symptoms	Suicidality
age	0.05 (0.07)	0.05 (0.07)	−0.02 (0.07)	−0.03 (0.11)
age^2^	−0.0003 (0.0007)	−0.0003 (0.0007)	0.0002 (0.0007)	0.001 (0.001)
female (vs. male)	0.37 (0.32)	0.35 (0.33)	0.28 (0.35)	0.35 (0.52)
married/cohabitating (vs. never married)	−0.95 (0.60)	−0.90 (0.63)	−0.59 (0.66)	−0.50 (1.10)
divorced/separated/widowed (vs. never married)	0.92 (0.88)	−1.24 (0.93)	−0.56 (1.06)	−1.54 (1.46)
college degree	0.24 (0.48)	0.25 (0.50)	0.17 (0.48)	0.28 (0.64)
chronic conditions	1.04 ** (0.32)	1.13 ** (0.32)	0.99 * (0.37)	0.83 (0.63)
friend support (*z*)	0.32 (0.17)	0.34 (0.18)	0.11 (0.24)	0.06 (0.26)
relative support (*z*)	−0.21 (0.18)	−0.22 (0.19)	−0.07 (0.18)	−0.0002 (0.32)
friend conflict (*z*)	0.47 (0.21)	0.49 (0.22)	0.005 (0.25)	0.47 (0.32)
relative conflict (*z*)	0.47 * (0.21)	0.49 * (0.22)	0.31 (0.26)	0.12 (0.41)
constant	−3.65 (1.57)	−3.63 (1.62)	−2.28 (1.33)	−3.33 (2.05)

*n* = 468; * *p* < 0.05, ** *p* < 0.01.

**Table 7 brainsci-08-00016-t007:** Weighted Logistic Regression on Four Dimensions of Depression among US-born Chinese Americans.

	Model 2-1	Model 2-2	Model 2-3	Model 2-4
	Negative Affect	Somatic Symptoms	Cognitive Symptoms	Suicidality
age	−0.18 (0.09)	−0.18 (0.09)	−0.11 (0.10)	0.02 (0.11)
age^2^	0.002 (0.001)	0.002 (0.001)	0.0003 (0.001)	−0.001 (0.001)
female	1.83 ** (0.52)	1.83 ** (0.52)	2.07 *** (0.48)	1.86 ** (0.52)
married/cohabitating (vs. never married)	−1.27 (1.19)	−1.27 (1.19)	−1.43 (1.39)	−1.43 (0.97)
divorced/separated/widowed (vs. never married)	1.81 (0.98)	1.81 (0.98)	2.23 (1.30)	0.97 (1.14)
college degree	1.35 ** (0.42)	1.35 ** (0.42)	0.98 * (0.45)	0.15 (0.61)
chronic conditions	1.32 * (0.53)	1.32 * (0.53)	1.27 * (0.70)	0.68 (0.75)
friend support (*z*)	−0.14 (0.31)	−0.14 (0.31)	−0.67 * (0.30)	−0.39 (0.38)
relative support (*z*)	0.01 (0.24)	0.01 (0.24)	0.07 (0.27)	0.28 (0.26)
friend conflict (*z*)	0.09 (0.29)	0.09 (0.29)	0.20 (0.34)	−0.45 (0.35)
relative conflict (*z*)	0.75 (0.36)	0.75 (0.36)	0.67 (0.40)	0.68 (0.37)
constant	0.59 (1.60)	0.59 (1.60)	−0.05 (1.86)	−2.65 (2.43)

*n* = 121; * *p* < 0.05, ** *p* < 0.01.

**Table 8 brainsci-08-00016-t008:** Weighted Logistic Regression on Four Dimensions of Depression among Non-Hispanic Whites.

	Model 3-1	Model 3-2	Model 3-3	Model 3-4
	Negative Affect	Somatic Symptoms	Cognitive Symptoms	Suicidality
age	0.10 *** (0.02)	0.10 *** (0.02)	0.11 *** (0.02)	0.10 *** (0.02)
age^2^	−0.001 *** (0.0002)	−0.001 *** (0.0002)	−0.001 *** (0.0002)	−0.001 *** (0.0002)
female	0.51 *** (0.09)	0.53 *** (0.08)	0.51 *** (0.09)	0.55 *** (0.08)
married/cohabitating (vs. never married)	−0.42 ** (0.14)	−0.41 ** (0.15)	−0.41 ** (0.14)	−0.46 ** (0.14)
divorced/separated/widowed (vs. never married)	0.20 (0.16)	0.21 (0.16)	0.22 (0.15)	0.13 (0.17)
college degree	−0.04 (0.09)	−0.03 (0.09)	−0.10 (0.08)	−0.14 (0.09)
chronic conditions	0.76 *** (0.08)	0.77 *** (0.09)	0.78 *** (0.09)	0.77 *** (0.11)
friend support (*z*)	−0.02 (0.06)	−0.02 (0.06)	−0.0002 (0.04)	−0.02 (0.06)
relative support (*z*)	−0.16 *** (0.04)	−0.16 *** (0.04)	−0.18 *** (0.04)	−0.22 *** (0.04)
friend conflict (*z*)	−0.002 (0.04)	−0.002 (0.04)	0.04 (0.05)	0.01 (0.05)
relative conflict (*z*)	0.21 *** (0.04)	0.21 *** (0.04)	0.21 *** (0.04)	0.23 *** (0.05)
constant	−3.20 (0.38)	−3.24 (0.37)	−3.46 (0.37)	−3.20 (0.38)

*n* = 3749; * *p* < 0.05, ** *p* < 0.01, *** *p* < 0.001.

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
