# Peer review of "Depression Symptom Patterns and Social Correlates among Chinese Americans"

_brainsci, 2018, doi:10.3390/brainsci8010016_

Round 1
Reviewer 1 Report
The introduction and discussion are strong, and the CPES is a validated study. Sample size is good, and the results are representative. The literature review can still be enhanced. For instance, please see DOI: 10.15171/ijer.2017.02 and doi: 10.1186/2251-6581-13-36. doi:10.3390/jpm7040006 and doi: 10.4103/2008-7802.164413. These studies all show that correlates of subjective health and mental well-being and psychiatric disorders vary across ethnic groups, and some of these studies have included people from China and United States. Please
These studies have very important implications by advocating for tailored interventions and policy making that are specific to the context and population.
Please justify why you did not use data from the other ethnic groups?
Socioeconomic factors differently influence health of ethnic groups. Use the above literature and justify the findings.
Author Response
Reviewer 1 Comments | Response |
The introduction and discussion are strong, and the CPES is a validated study. Sample size is good, and the results are representative. The literature review can still be enhanced. For instance, please see DOI: 10.15171/ijer.2017.02 and doi: 10.1186/2251-6581-13-36. doi:10.3390/jpm7040006 and doi: 10.4103/2008-7802.164413. These studies all show that correlates of subjective health and mental well-being and psychiatric disorders vary across ethnic groups, and some of these studies have included people from China and United States. Please | A few more recent articles are added, including a couple of relevant articles suggested by the reviewer. |
Please justify why you did not use data from the other ethnic groups?
| A brief justification is provided in the literature review section, i.e., the heterogeneity of depression prevalence and pattern between Asian groups.
|
Socioeconomic factors differently influence health of ethnic groups. Use the above literature and justify the findings.
| A section on sociodemographics is added in the literature review to provide justification. Relevant literature was cited. |
In addition, the author proofread the manuscript, further edited the in-text citations, and corrected typos.
Reviewer 2 Report
Abstract.
1st paragraph, 1st sentence: Replace goal with aim. The abstract is unclear and vague in sections, particularly with social correlates and the four dimensions. The abstract would improve if you identify the social correlates in your abstract as well as the four dimensions. I am guessing that the factor analysis identified the four dimensions?
Introduction
Your Introduction is very interesting and raises very valid points to consider. The cultural aspects of psychiatric symptoms are often overlooked as they are considered through a Western lens. I think that your Introduction would be strengthened and more comprehensive if you gave a better description of the dimensions of depression as well as Chinese vs Western manifestations of depression. See further comments below.
First paragraph could give examples (in one sentence) of how Chinese Americans manifest depression differently.
The first two paragraphs could be worded more clearly to describe how Chinese Americans differ from Western American society. For example, the first sentence in the second paragraph (starting line 36) could explain how Western American society is different to Chinese holism.
First sentence (line 43) of the third paragraph does not describe how depression is multidimensional.
Somatic symptoms subsection: The following sentence (line 70-71) is tautological. “The term somatization is used to refer to the reporting of somatic symptoms, especially medically unexplained somatic symptoms by patients with psychiatric disorders [19” Your sentence would be clearer if you replace “reporting of somatic symptoms” with reporting of symptoms such as …….”.
The affective symptoms subsection does not really describe depressive affect and seems to be an extension of the (above) somatic symptoms subsection.
Other depression dimensions only describes other studies of Asian vs White populations. Does not discuss dimensions other than a discussion of somatisation vs non-somatisation symptoms of cognition and interpersonal conflict. Your last sentence in this paragraph is clear and gives a better description.
There is no paragraph summarising the aims and hypotheses of this study. Please write this.
Methods
Data Source subsection: Line 107. “This study USED”. Must use past tense in academic articles.
Make sure that writing is in past tense.
2.2.1 DSM IV assessment section does not name an instrument or reference the authors of that tool. Please fix. Same for 2.2.4, social supports.
2.3 Statistical analysis subsection (line175-177): “I conduct exploratory factor analysis on all twenty-four items of MDD symptoms, to examine the patterns of depression symptoms for three groups, foreign-born Chinese Americans, US-born Chinese Americans, and non-Hispanic whites.” Re-word to remove “I conduct”. Such as “Exploratory factor analysis was conducted on all….”.
2.3.1. Descriptive statistics subsection to be mentioned first? (I.e., before the factor analysis?). Remove the “I examined” as per 2.3 comment above.
Re-word any “I examined” in the rest of the Methods and Results section.
You need to consider if the factor analysis and demographics should be in the Results section of this paper.
Results
In the first paragraph of the Results section, the four dimension need to be named again.
Many tables and associations for the reader to consider. The changes to the Introduction about the depression dimensions, social constructs and cultural differences would help the reader with the depth of comparisons in the Results section. Having clear aims for you study would further clarify what your results are exploring and the story that they are telling.
Discussion
First paragraph (line 341-342): “In other words, for all three groups, these four items these four items seem to form a distinct construct.” This sentence needs editing and is unclear in meaning.
The Discussion would be improved by having clear study aims and then re-format the Discussion with paragraphs summarising the answers/findings to those study aims.
The second paragraph should focus on DSM 5 instead of DSM-IV.
The third paragraph does not mention the social cultural factors that were identified in the Results section that mental health clinicians need to consider for each cultural group.
Limitations section is good. Other limitation are that the surveys use DSM-IV criteria instead of DSM 5. I am unsure if the measures are appropriate as they were not named in the Methods section.
The second last sentence is very unclear and convoluted (Lines 382-385). Be specific on what is recommended for future research.
“Findings of the present study provide suggestions for future research. Cultural norms regarding depression and social stigma have been theorized and hypothesized to affect the expression of depressive symptoms, yet there have been very few empirical studies that test such effects, largely due to the difficulty in operationalizing stigma and cultural norm. Findings from previous research [55,56] shed light on this topic, which merit future research.”
The conclusion could specifically say how the three cultural groups differ in depression symptom patterns and how social factors affect them. Current sentences/statements are vague and water down the importance of the study findings.
Author Response
Reviewer's comments are most sincerely appreciated. Addressing the comments really help me thinking more deeply and clearly about this study, and my future research in this topic.
